# GENERATION BY SEARCH: SCALING TEST-TIME COMPUTE FOR AUTOREGRESSIVE IMAGE GENERATION

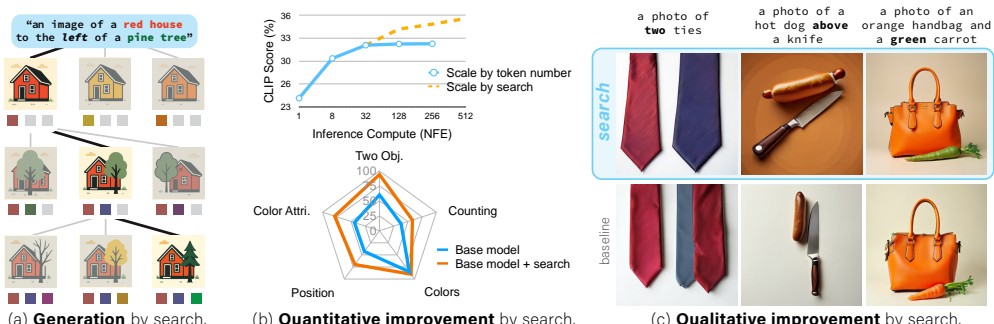

(a) **Generation** by search.  (b) **Quantitative improvement** by search.  (c) **Qualitative improvement** by search.

Figure 1: (a) **Main concept:** We formulate image generation as a search problem, where we search over possible token sequences to maximize certain task utilities (often provided by a "verifier"). (b) **Quantitative gain:** Search adds desirable test-time scaling behavior to a base model (here, FlexTok (Bachmann et al., 2025)). It improves the image-prompt alignment on the image generation benchmark GenEval by a large margin (shown on Janus model (Wu et al., 2024)). (c) **Qualitative gain:** The characters of the improvements yielded through search demonstrated by a few examples. Difficult GenEval (Ghosh et al. (2023)) prompts can be solved through search.

## ABSTRACT

Image generation has made significant progress in recent years, but still faces difficulties in following challenging prompts and poor scalability with respect to inference-time computation. In this paper, we propose framing autoregressive image generation as a search problem, where the objective is to identify token sequences that maximize a chosen utility function at the test time. This framework enables fine-grained control over the generation process through flexible choices of utility functions and yields better scaling behavior as more test-time compute is used. Moreover, it is fully compatible with existing autoregressive generative models, which can be viewed as providing token-level priors during the search. To systematically investigate this framework, we organize the design space into four key axes and conduct studies across the choices of *token structure* (2D grid, 2D multi-scale, and 1D ordered), *search algorithm* (best-of-N, beam search, and lookahead search), *verifier* (optimizing for image-text and image-image alignment, as well as quality), and *prior model* (conditional and unconditional autoregressive models, and prior-free). Together, these findings establish search as a performant, controllable, and scalable approach to advancing image generation and provide practical guidance for future work in this direction.

## 1 INTRODUCTION

Generative models have advanced rapidly in recent years, driving breakthroughs in vision (Ramesh et al., 2021; Rombach et al., 2022) and language (Brown et al., 2020; Touvron et al., 2023) tasks. A simple and widely used approach is autoregressive (AR) modeling, where a model is trained to predict the next token given the previous ones. In language, AR prediction of text tokens (Radford & Narasimhan, 2018) forms the basis of many large language models (LLMs), while in vision, AR

models trained on image tokens (van den Oord et al., 2017; Esser et al., 2020) have demonstrated strong generative performance (Yu et al., 2022; Sun et al., 2024; Han et al., 2025; Bachmann et al., 2025) on par with other generative model classes like diffusion models (Rombach et al., 2022; Peebles & Xie, 2023). AR Transformers scale well across both domains, improving consistently with increased compute, larger datasets, and model size (Kaplan et al., 2020; Hoffmann et al., 2022; Henighan et al., 2020; Yu et al., 2022; Tian et al., 2024).

Beyond scaling training compute, recent work on LLMs show that scaling *test-time compute* can significantly improve performance, especially on challenging reasoning tasks like mathematics and planning (Brown et al., 2024; Snell et al., 2024). Instead of directly producing a single answer, models can spend extra computation exploring potential solutions and performing structured search guided by verifiers. Naturally, this raises the question: *Can image generation also benefit from scaling test-time compute?*

In this paper, we propose a search-based test-time scaling framework for token-based image generation, which we call *Search over Tokens*, in short `SoT`. It broadly enhances AR image model performance during inference, without retraining or finetuning. To achieve this, we reformulate image generation as a search problem (Fig. 1, left): each partial sequence of tokens defines a state in a search tree, each possible next token is an action, and a pre-trained AR model serves as a prior that guides the search by pruning unlikely continuations. The goal is to find a complete sequence of tokens that maximizes a verifier score (e.g., image-text alignment, aesthetics, spatial layout, etc.). Fig. 2 provides an overview of the search framework, compared with standard AR inference.

Building on this formulation, we identify four critical axes that determine the design and effectiveness of the search process: (1) the *token structure*, which shapes how the state space is represented and expanded; (2) the *search algorithm*, which determines how the space is explored; (3) the *verifier*, which determines the search objective; and (4) the *next-token prior*, which provides next-token probabilities to narrow down the search space. Specifically, we explore three common token structures: 1D ordered token sequences as in FlexTok (Bachmann et al., 2025), 2D multi-scale tokens as in VAR (Tian et al., 2024; Han et al., 2025), and 2D grid tokens as in Janus (Wu et al., 2024) and Janus-Pro (Chen et al., 2025a). For each token structure, we study three search algorithms: best-of-N sampling, beam search, and look-ahead search. We also examine a wide variety of verifiers, including CLIPScore (Radford et al., 2021; Hessel et al., 2021), aesthetic score (Schuhmann et al., 2022), ImageReward (Xu et al., 2023), rule-based verifiers (Ghosh et al., 2023), and vision-language models (Bai et al., 2025; Laurençon et al., 2024). Finally, we study the role of the AR prior, including conditional and unconditional AR models as well as the extreme case where no AR model is available.

We conduct comprehensive experiments to study the proposed framework, evaluating it on a variety of text-to-image benchmarks including GenEval (Ghosh et al., 2023), COCO (Lin et al., 2014), Dreambench++ (Peng et al., 2024), DrawBench (Saharia et al., 2022) and DPG Bench (Hu et al., 2024). Our experiments reveal several key findings: (1) Test-time scaling through search significantly and consistently improves performance (see Fig. 1) across a wide range of AR models and token structures. For example, on the GenEval benchmark, all studied models achieve 10%-20% average accuracy improvements with test-time search. (2) Test-time scaling compensates for limited train-time scaling. For example, even with moderate test-time scaling, a 530M parameter AR model outperforms a 3.4B model, but the larger model benefits more from further scaling inference compute. (3) Using different verifiers allows us to generate images guided towards distinct preferences, and supports multimodal control without additional training. (4) The token structure influences the search algorithm choice: AR models trained on ordered tokens benefit more from tree-based search algorithms, while models operating on 2D grid tokenizers perform better with best-of-N sampling. (5) Interestingly, using an ordered token structure can eliminate the need for an AR model altogether, enabling generation directly through search over tokens.

We summarize our contributions as follows:

1. We propose a framework that enhances AR image generation models without retraining or finetuning, by casting it as a search problem guided by verifiers.

2. We identify four critical axes of test-time search: token structure, search algorithm, verifier, and AR model, and systematically study their roles and interactions.

Figure 2: **Overview of our SoT framework. Left:** The AR model outputs a single next token at each step without search. **Right:** Our SoT framework explores multiple candidate tokens, detokenizes them into images, and leverages a verifier to select the best tokens, enabling more controlled and task-aligned generation.

3. Through a diverse set of experiments, we demonstrate the remarkable benefits of test-time scaling via search and provide practical guidance for future research on effective design choices.

## 2 RELATED WORK

**AR image generation.** AR transformers model data as sequences, predicting each token based on those generated earlier. In language, this next-token prediction framework has become the cornerstone of large-scale models such as GPT (Achiam et al., 2023). Extending this idea to vision, early approaches like iGPT (Chen et al., 2020) treated images as flattened pixel sequences, which proved effective for small images but limited in scalability and quality. Recent progress has shifted focus toward designing structured visual tokens and generation orders, as the organization of tokens plays a central role in both model efficiency and visual fidelity. LlamaGen (Sun et al., 2024) leverages spatially aligned two-dimensional grid tokens to reach image quality on par with diffusion models, while VAR (Tian et al., 2024) introduces a multi-scale generation order that predicts images progressively from coarse to fine details. These developments illustrate a spectrum of token designs, ranging from strictly one-dimensional sequences such as FlexTok (Bachmann et al., 2025) and Selftok (Wang et al., 2025), to two-dimensional ordered layouts such as VAR (Tian et al., 2024) and Infinity (Han et al., 2025), to fully grid-aligned tokens as used in Janus (Wu et al., 2024) and LlamaGen (Sun et al., 2024).

**Search for image generation.** Search-based methods improve image generation by exploring multiple generation trajectories at inference time. In diffusion models, search typically operates over the continuous noise space to refine generation quality or increase diversity (Ma et al., 2025; Singhal et al., 2025; Zhang et al., 2025). For example, Ma et al. (2025) introduces path-based search strategies that explore alternative generation trajectories and leverage a verifier to improve alignment with the target objective. While these methods focus on continuous noise, AR models enable a different form of search by directly operating over discrete token sequences. This approach is naturally compatible with modern LLMs, and it offers advantages such as finer control, easier integration with multimodal tasks, and direct token-level reasoning. Prior works like TTS-VAR (Chen et al., 2025b) have explored search within specific AR architectures. Here, we study token-level search more broadly, highlighting its potential as a general and flexible framework for improving image generation, parallel to noise-based search in diffusion models.

Please refer to Appendix A for additional, more detailed discussion of the relevant work.

## 3 METHOD

### 3.1 PRELIMINARIES AND PROBLEM FORMULATION

**Background on autoregressive image generation.** Consider an image represented as a sequence of $T$ discrete tokens $\mathbf{x} = (x_1, x_2, \ldots, x_T)$, where each $x_t \in V$ is an index from a vocabulary $V$. These tokens are usually the output of a discrete tokenizer (van den Oord et al., 2017; Esser

et al., 2020) encoder. An AR model estimates the probability of the full sequence by factorizing it into conditional distributions over successive tokens, i.e., $p(\mathbf{x}) = \prod_{t=1}^{T} p(x_t \mid x_{<t})$, with $x_{<t} = (x_1, \ldots, x_{t-1})$ denoting the prefix up to step $t-1$. During inference, tokens are generated one by one, where each $x_t$ is sampled or chosen conditioned on the previously generated prefix, until the entire sequence is completed.

**Formulating image generation as a search problem.** Generating discrete tokens one-by-one can be interpreted as navigating a search tree, with every level corresponding to the decision which next token index to add to the sequence. Concretely, at step $t$, the state is defined as $s_t = (x_1, \ldots, x_t)$, i.e., the prefix of the token sequence up to position $t$. An action corresponds to selecting the next token $x_{t+1} \in V$ from the vocabulary, which transitions the state to $s_{t+1} = (x_1, \ldots, x_t, x_{t+1})$. The AR model provides a prior probability distribution over possible actions $p(x_{t+1} \mid x_{\leq t})$. The search begins from the initial state $s_0 = \varnothing$ (or a conditioning prompt) and terminates when a complete sequence $s_T = \mathbf{x}$ is formed. The objective is to find a terminal state $s_T$ that maximizes the likelihood or a task-specific utility, typically defined by a verifier function $\mathcal{G}$. Figure 2 shows the framework.

In this formulation, standard AR generation performs a greedy search on each token based on the predicted next-token probabilities. However, such greedy decoding does not necessarily lead to the highest likelihood of the full sequence. Moreover, token sequences with a high likelihood do not necessarily correspond to images that align well with different downstream objectives, such as image preference or condition alignment.

### 3.2    FOUR KEY COMPONENTS FOR SEARCH OVER TOKENS.

Under the test-time search framework, we address these issues by: (1) exploring more sophisticated *search algorithms* to better optimize the objective, and (2) using a ***verifier*** model to directly measure expected task utility, such as image–text alignment, to guide generation. Besides, (3) the ***token structure*** (how the state space is represented and expanded), and (4) the ***AR model*** (the prior model's next-token probabilities) also play a central role in the search framework and may affect the choice for different search algorithms. We discuss these four key components for search in detail in the following.

**Token structure.** Image tokenization plays a critical role in autoregressive image generation, as it defines the basic components used to factorize the joint distribution. Common approaches tokenize images into a fixed 2D grid of tokens, i.e., clusters of tokens encode regions of the image. AR models are often traine to predict these token grids in raster order. Recent advances in image tokenization (Miwa et al., 2025; Bachmann et al., 2025; Wen et al., 2025) represent an image as an ordered sequence of tokens, typically progressing from coarse to fine. AR models trained on this representation learn to generate images from coarse to fine, with the process better aligned with human perception.

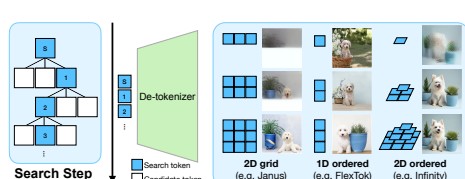

Figure 3: **Token structure in search.** SoT explores candidate tokens, which are then detokenized into images. Janus employs fixed 2D grids, FlexTok uses coarse-to-fine ordering, and Infinity utilizes multi-scale 2D ordered tokens.

Different token structures decide different search states and how each state transitions to the next, and thus may strongly affect the search results. To provide an overview that covers various token types, we consider Janus and Janus Pro, state-of-the-art models for autoregressive image generation with 2D grid tokens, as well as FlexTok, which introduces an ordered tokenization that represents images from coarse to fine, and Infinity, which use 2D ordered tokens. We show an example of different token structures in Figure 2.

**Search algorithm.** The search space in token-based image generation is large. For a vocabulary size of $|V|$ and number of tokens $T$, the total search space is $|V|^T$. In modern AR models, $|V|$ is typically on the order of $10{,}000$, and $T$ is on the order of $100$. This makes the full search space computationally intractable to explore directly. We therefore need to carefully choose search algorithms

that can effectively navigate this space. Specifically, we consider three algorithms: Best-of-$N$ sampling, beam search, and lookahead search. These cover different assumptions about structure and the need for lookahead. We explain each search method and its properties in detail in App. B.1.

**Verifier.** Verifiers serve as the search objective and guide the generation process by evaluating the utility of partial token sequences. We explore a broad set of verifiers to examine what can be achieved with different objectives, and what characteristics make a good verifier. We consider three main objectives to optimize over, namely (1) image-text alignment (e.g., using CLIPScore), (2) overall image quality (e.g., aesthetics), and (3) image-image alignment (e.g., with DreamSim). We further find that optimizing for individual verifiers tends to overfit on them, at the expense of the performance in other evaluation domains. To that end, we explore ensembles of verifiers and show that they are more robust to overfitting. We discuss the different verifier choices in detail in App. B.2.

**Autoregressive prior.** Lastly, we consider the role of AR models, which provide prior probabilities for the next token and help prune the search space to its most likely regions. However, from the perspective of search, this component is not strictly required. We therefore ask: is the AR model essential? To investigate, we first replace the AR model with an unconditional probability and then remove it entirely. Surprisingly, we find that with the ordered tokenization scheme in FlexTok, one can either use an unconditional AR model or even completely ignore AR models to achieve reasonable text-to-image generation performance.

## 4 EXPERIMENTS

We evaluate the effectiveness of SoT across multiple image generation settings. Our experiments are designed to answer three key questions: (1) How much can search improve autoregressive (AR) generation models, and does this generalize across token structures? (2) Can search enable new capabilities, such as multimodal control, without additional training? (3) What design choices across search algorithms, token structure, verifier selection, and AR prior models, lead to the most effective search strategies? We first present the main results on standard benchmarks, then analyze the impact of different design axes, and finally study the tradeoff between training-time and test-time compute.

### 4.1 MAIN RESULTS

**Search improves condition alignment across AR models.** We begin by investigating whether search algorithms benefit AR generation models, and to what extent. To this end, we conduct experiments on four AR generation models: FlexTok (Bachmann et al., 2025), Infinity (Han et al., 2025), Janus (Wu et al., 2024), and Janus-Pro (Chen et al., 2025a), and evaluate on the GenEval Ghosh et al. (2023) benchmark. For each model, we explore best-of-N sampling ($N = 50$) and a tree search algorithm that best leverages the structure of different tokenizers. Specifically, we adopt beam search with lookahead on models that operate on 2D tokens, namely Janus, Janus-Pro, and Infinity. On FlexTok's ordered 1D tokens, we find that beam search alone already achieves strong results while being more computationally efficient than lookahead search (see Figure 6). We use *ImageReward* as the verifier. We provide further implementation details in Appendix B and C, and an ablation study on different lookahead numbers and additional hyperparameter tuning results in Appendix D.

As shown in Table 1, both tree search and best-of-$N$ search consistently improve the AR baselines (FlexTok, Infinity, Janus, and Janus-Pro) by a substantial margin (often by up to 10-30 percentage points). Notably, for challenging settings like GenEval's *two objects* and *position* categories, our method improves Janus and FlexTok by about 30 and 10 percentage points, respectively. These results indicate that search serves as an effective approach to improve image-text alignment. We show a visual comparison of search vs. no-search in Fig. 4. We also evaluate SoT with long prompts from the DPG benchmark (Hu et al., 2024), and show the results in the Appendix in Fig. 11.

We further compare the highest-performing AR model with SoT (Janus-Pro with lookahead search) to other generative models without search. As shown in Tab. 2, our method achieves strong performance on the GenEval benchmark, outperforming competitive baselines.

Table 1: **SoT consistently improves performance of different AR models on GenEval.** The second column specifies the Search over Tokens (SoT) type: a dash (–) for vanilla models, Best-of-$N$ (BoN, N=50), Lookahead (LA), or Beam search. Increment values indicate absolute gains over each model's baseline. Notably, for challenging settings of two objects, counting, and color attributes, search improves performance by around 10 to 30 percentage points over the base model.

| Model | Search | Single obj. | Two obj. | Counting | Colors | Position | Color attri. | Overall ↑ |
|---|---|---|---|---|---|---|---|---|
| FlexTok | – | 95 | 59 | 56 | 80 | 16 | 35 | 57 |
| (Bachmann et al., 2025) | BoN | 100 +5 | 84 +25 | 69 +13 | 90 +10 | 24 +8 | 57 +22 | 68 +11 |
| | Beam | 100 +5 | 88 +29 | 69 +13 | 91 +11 | 23 +7 | 53 +18 | 70 +13 |
| Infinity | – | 98 | 82 | 65 | 83 | 27 | 64 | 70 |
| (Han et al., 2025) | BoN | 100 +2 | 93 +11 | 71 +6 | 83 +0 | 30 +3 | 67 +3 | 74 +4 |
| | LA | 100 +2 | 93 +11 | 69 +4 | 91 +8 | 36 +9 | 74 +10 | 77 +7 |
| Janus | – | 96 | 60 | 38 | 85 | 43 | 44 | 61 |
| (Wu et al., 2024) | BoN | 96 +0 | 91 +31 | 51 +13 | 90 +5 | 65 +22 | 55 +11 | 75 +14 |
| | LA | 100 +4 | 94 +34 | 58 +20 | 90 +5 | 70 +27 | 79 +35 | 82 +21 |
| Janus-Pro | – | 100 | 86 | 60 | 91 | 76 | 60 | 79 |
| (Chen et al., 2025a) | BoN | 97 -3 | 91 +5 | 74 +14 | 90 -1 | 77 +1 | 78 +18 | 85 +6 |
| | LA | 100 +0 | 95 +9 | 76 +16 | 94 +3 | 81 +5 | 79 +19 | 87 +8 |

Table 2: **System-level comparison on GenEval.** We apply SoT to the Janus-Pro model with the ImageReward verifier and compare it with other existing methods. Models marked with † are reproduced by us for evaluation. SoT achieves the best performance on GenEval.

| Model | Single obj. | Two obj. | Counting | Colors | Position | Color attri. | Overall ↑ |
|---|---|---|---|---|---|---|---|
| *Models without test-time search* | | | | | | | |
| CLIP Retrieval Beaumont (2022) | 89 | 22 | 37 | 62 | 3 | 0 | 35 |
| SD-XL (Podell et al., 2023) | 98 | 74 | 39 | 85 | 15 | 23 | 55 |
| LlamaGen (Sun et al., 2024) | 75 | 26 | 20 | 55 | 42 | 32 | 31 |
| LlamaGen-GRPO (Yuan et al., 2025) | 79 | 26 | 23 | 59 | 40 | 30 | 32 |
| Emu3-Gen Wang et al. (2024) | 98 | 71 | 34 | 81 | 17 | 21 | 54 |
| FlexTok† (Bachmann et al., 2025) | 95 | 59 | 56 | 80 | 16 | 35 | 57 |
| Janus† (Wu et al., 2024) | 96 | 60 | 38 | 85 | 43 | 44 | 61 |
| Show-o (Xie et al., 2024) | 98 | 80 | 66 | 84 | 31 | 50 | 68 |
| Infinity† (Han et al., 2025) | 98 | 82 | 65 | 83 | 27 | 64 | 70 |
| Janus-Pro† (Chen et al., 2025a) | 100 | 86 | 60 | 91 | 76 | 60 | 79 |
| GPT-4o-Image (Yan et al., 2025) | 99 | 92 | **85** | 89 | 74 | 71 | 84 |
| *Models with test-time search* | | | | | | | |
| TTS-VAR (Chen et al., 2025b) | – | 95 | 74 | – | – | 68 | 75 |
| **SoT-Janus-Pro (Ours)** | **100** | **95** | 76 | **94** | **81** | **79** | **87** |

**Search enables zero-shot multimodal control.** Besides image-text alignment, we also find that search enables conditioning the generation on modalities that have not been trained on in the base model. Here, we explore the task of generating an image from a text prompt, while preserving a concept from a given reference image. We do this by using an image-image similarity verifier, here, *DreamSim* (Fu et al., 2023), between the generated images and the reference images. We use the FlexTok model with beam search and evaluate on a concept preservation benchmark, Dream-Bench++ (Peng et al., 2024). Results in Figure 5 show that SoT improves concept preservation around 30% for DINO-I score and 10% on CLIP score, while maintaining good prompt following performance.

## 4.2 UNDERSTANDING THE SoT DESIGN SPACE

To better understand how to effectively apply search on top of various AR models and guide them towards different objectives, we analyze four main design axes, namely the token structure, the choice of search algorithm, the verifiers, and the use of AR priors.

**Token structure and search algorithms.** The choice of token structure influences the optimal choice of search algorithm to apply. For example, 1D ordered tokenizers like FlexTok (Bachmann et al., 2025) structure tokens in a coarse-to-fine manner, and enable decoding truncated token se-

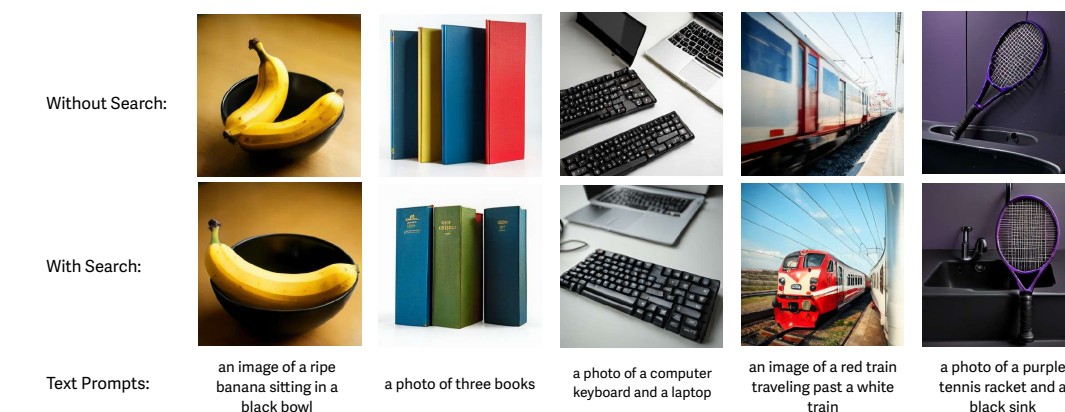

|  |  |  |  |  |  |
| --- | --- | --- | --- | --- | --- |
| Without Search: | | | | | |
| With Search: | | | | | |
| Text Prompts: | an image of a ripe banana sitting in a black bowl | a photo of three books | a photo of a computer keyboard and a laptop | an image of a red train traveling past a white train | a photo of a purple tennis racket and a black sink |

Figure 4: **Visualization of generation with SoT.** We show the visual comparison of Janus-Pro and Janus-Pro+SoT (Ours) using prompts from the GenEval and COCO datasets and ImageReward as the verifier. Search improves image–text alignment and enhances control over the generation.

quences into valid images. In contrast, 2D grid tokenizers (Sun et al., 2024), as used by Janus and Janus-Pro, require completing the entire token sequence to decode an image, either by AR generation or padding.

To examine the behavior of search algorithms on differen token structures, we conduct an apples-to-apples comparison between AR models trained on 1D ordered token sequences (FlexTok), and 2D grid tokens trained in a compute- and data-controlled manner (Bachmann et al., 2025). Specifically, we compare both token structures using best-of-N sampling, beam search, and lookahead search. For best-of-N, we scale N up to 500. For beam search and lookahead search, we vary the number of searched tokens from 1 to 256. All experiments are conducted on a subset of the COCO Karpathy validation set (Lin et al., 2014) containing 300 images. We measure the scaling trends against the compute spent in terms of Number of Function Evaluations (NFEs), where we count generating one token or verification as one function evaluation. Please see Appendix C for details of NFE.

As shown in Figure 6, both best-of-N search and lookahead search exhibit similar scaling trends across the two models. In contrast, beam search performs better with 1D-ordered tokens. We attribute this to the fact that a partially decoded 1D-ordered token sequence carries much richer information than a partial 2D-grid token sequence. As a result, during intermediate search steps, 1D-ordered tokens produce more informative detokenized images, which in turn provide stronger and more accurate verifier guidance. For best-of-N and lookahead search, however, the difference is less pronounced because both methods require rolling out to the end of the sequence, making the role of intermediate tokens less significant. We also compare the best-of-N search for AR generation and for diffusion models on the DrawBench in the Appendix F.1.

**Verifiers.** We study the impact of different verifiers used to guide the search. To this end, we perform a leave-one-out evaluation on seven verifiers: CLIPScore (Radford et al., 2021; Hessel et al., 2021), aesthetic score (Schuhmann et al., 2022), Cyclereward (Bahng et al., 2025),HPSv2 (Wu et al., 2023), ImageReward (Xu et al., 2023), Grounded SAM (Ghosh et al., 2023), PickScore (Kirstain et al., 2023), as well as an ensemble of them. We evaluate the results on GenEval using FlexTok with beam search.

As shown in Fig. 7, each verifier performs best under its own evaluation metric, but often performance degrades on held-out metrics. This suggests that these verifiers are not robust enough, and that the search is performing verifier hacking. That said, the ensemble verifier typically ranks second across individual metrics and achieves the best average ranking, meaning it can serve as a more robust overall verifier by optimizing for a broader range of objectives. In addition, ImageReward and HPSv2 achieve the next-best average rankings, indicating that human-preference models can also be a robust choice when multiple evaluations are taken into account.

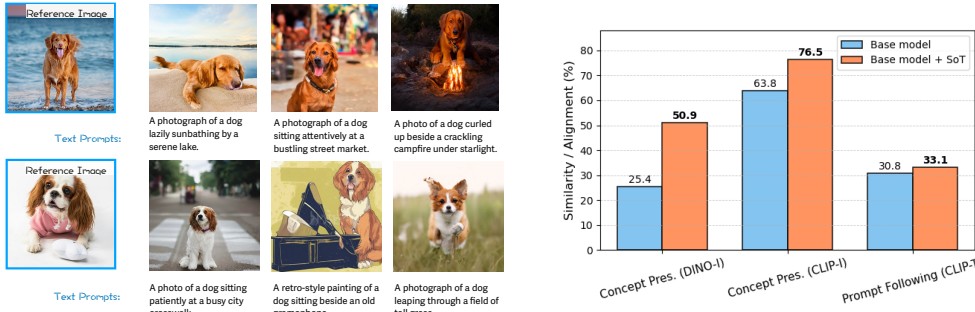

Figure 5: **Image generation with concept preservation through search.** `SoT` enables multimodal generation control without any finetuning, purely through applying search with an image similarity verifier, here, *DreamSim* (Fu et al., 2023). **(Left):** Example outputs produced by `SoT`-FlexTok for different reference images and prompts. **(Right):** Quantitative results on DreamBench++ (Peng et al., 2024), evaluating concept preservation and text alignment. We compare the no-search results of the FlexTok AR model with those obtained using `SoT` (with beam search).

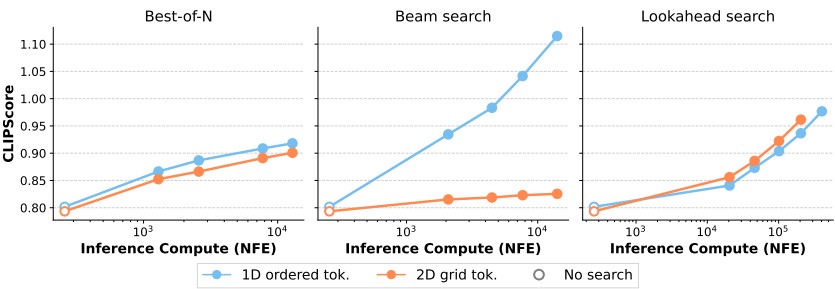

Figure 6: **Search algorithm comparison on two token structures.** We conduct apples-to-apples experiments using different search algorithms on two tokenizers: 1D ordered (FlexTok) and a controlled 2D grid tokenizer variant. Our results show that there is little difference between the two when using best-of-N or lookahead search, but FlexTok benefits more from beam search. NFE = Number of Function Evaluations. Results are evaluated on COCO (Lin et al., 2014).

**AR prior models.** Lastly, we examine the role of the prior model during search. The main utility of the AR prior is to propose high likelihood token candidates, which drastically reduces the search space that needs to be verified. That said, the coarse-to-fine structure of certain tokenizers like FlexTok makes them amenable to explore the question: Is it possible to generate images without any AR prior, and purely through search? To do so, we perform beam search with a uniform prior, which assigns the same probability to each candidate token. In addition, we evaluate an unconditional prior model, where we use the FlexTok AR model without text conditioning. We find that search without an AR prior is indeed possible and yields reasonable results (Fig. 14, top row). Both unconditional and conditional priors, however, effectively narrow the search space towards more likely next tokens, enabling more efficient exploration. Please see Appendix H for full results.

### 4.3 TEST-TIME SCALING FOR DIFFERENT MODEL SIZES.

Lastly, we ask whether test-time compute can compensate for some training-time compute. To investigate this, we evaluate FlexTok AR on GenEval with different parameter sizes and find that a 530M parameter model with more test-time compute can actually outperform a larger 3.4B model with less test-time compute (Fig. 8). However, the larger model scales more effectively as inference compute increases. Overall accuracy traces a Pareto frontier against inference FLOPs, with the optimal model size rising alongside the compute budget and following a power-law scaling relation.

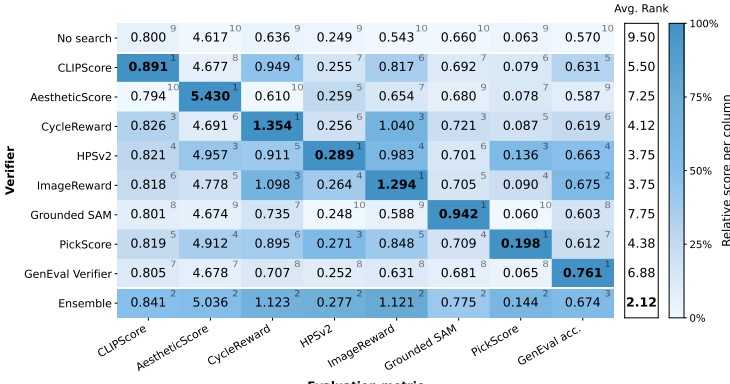

Figure 7: **Comparison of different verifiers.** Each row reports search using one verifier. All methods use the same beam search algorithm on FlexTok. The best score in each column is highlighted in bold. The superscript in each cell represents the rank within that column's metric, and the last column reports the average of these column-wise ranks, providing an overall rank for each verifier.

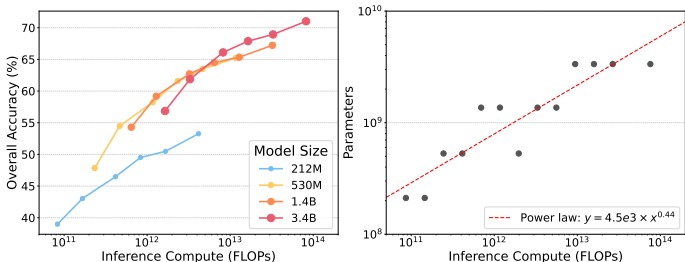

Figure 8: **Performance of search across different model sizes.** We study test-time scaling with different FlexTok AR sizes. **(Left):** We use GenEval with Best of N search and estimate the corresponding inference FLOPS. A small model with test-time search can be better than a large model directly doing inference. **(Right):** Extracting the model size with best GenEval performance within equally log spaced FLOPs buckets we find alignment with a power law relationship. Fitting a power law of the form $y = a \times x^b$ for the optimal model size as a function of inference compute, we find $a = 4.5 \times 10^3$ and $b = 0.44$.

## 5 DISCUSSION & CONCLUSION

In this work, we frame image generation as a search-over-tokens problem. We show the efficacy of using search as a test-time scaling method, consistently improving various pre-trained AR models in both generation quality and steerability. We also study four design axes and discuss the impact of token structure, search algorithms, verifiers, and prior models. Below, we discuss some limitations and directions for future work.

- *Verifier quality.* As search optimizes against a chosen verifier, results depend heavily on verifier quality. Weak or biased verifiers can lead to 'verifier hacking', where the search exploits flaws in the scoring function. While our experiments show that current verifiers are not fully robust, they still provide useful guidance and enable better control; improving verifier robustness remains an important direction for future work.

- *Adaptive search.* We studied search under a fixed compute budget. A promising direction is to allow the model to learn when to stop searching. For example, the model could automatically allocate more compute for a complex prompt, while simpler prompts might require less. A principled approach to adaptive compute allocation could lead to more efficient search.

- *Search in other domains.* While we focus on image generation, our framework can be extended to other multimodal domains such as video generation or agentic applications as exciting future directions.

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

# APPENDIX

## TABLE OF CONTENTS

# A  RELATED WORK

**Test-time scaling and search in language models.**  Test-time scaling has emerged as a powerful complement to training-time scaling, improving model performance by allocating additional computation during inference rather than increasing model size or dataset scale. In large language models (LLMs), this has been explored through two main directions. The first is reasoning, where structured intermediate steps such as Chain-of-Thought (Wei et al., 2022) are generated to enhance interpretability and reliability. The second is search, which has become one of the most prominent forms of test-time scaling by exploring multiple reasoning paths and selecting the most promising output.

Search has a long history in AI, from classical algorithms like Dijkstra, A*, and MCTS to game-playing systems such as AlphaZero (Silver et al., 2018), where learned priors guide tree exploration to achieve superhuman performance. Similar ideas have recently emerged in large language models (LLMs), where test-time scaling improves performance by allocating extra computation during inference. Early decoding methods such as greedy search, beam search, and nucleus sampling treat generation as a single trajectory, while recent approaches expand this into a structured search space guided by verifiers, as seen in works like Let's Verify Step by Step (Lightman et al., 2023), Tree-of-Thought (Yao et al., 2023), and MCTS-based reasoning.

These advances show that search combined with strong generative priors can significantly enhance model capabilities. However, extending this principle to image generation is non-trivial: the search space is vastly higher dimensional and requires maintaining spatial consistency across tokens. Our work addresses these unique challenges by developing a token-level search framework tailored to autoregressive image models, bringing the benefits of test-time scaling to a new and complex domain.

**Autoregressive image generation.**  Autoregressive transformers model data as sequences, predicting each token based on those generated earlier. In language, this next-token prediction framework has become the cornerstone of large-scale models such as GPT (Achiam et al., 2023). Extending this idea to vision, early approaches like iGPT (Chen et al., 2020) treated images as flattened pixel sequences, which proved effective for small images but limited in scalability and quality. Recent progress has shifted focus toward designing structured visual tokens and generation orders, as the organization of tokens plays a central role in both model efficiency and visual fidelity. LlamaGen (Sun et al., 2024) leverages spatially aligned two-dimensional grid tokens to reach image quality on par with diffusion models, while VAR (Tian et al., 2024) introduces a multi-scale generation order that predicts images progressively from coarse to fine details. These developments illustrate a spectrum of token designs, ranging from strictly one-dimensional sequences such as FlexTok (Bachmann et al., 2025) and Selftok (Wang et al., 2025), to two-dimensional ordered layouts such as VAR (Tian et al., 2024) and Infinity (Han et al., 2025), to fully grid-aligned tokens as used in Janus (Wu et al., 2024) and LlamaGen (Sun et al., 2024), underscoring how token structure fundamentally shapes modern autoregressive image generation.

**Search for image generation.**  Search-based methods improve image generation by exploring multiple generation trajectories at inference time. In diffusion models, search (Ma et al., 2025; Singhal et al., 2025; Zhang et al., 2025) typically operates over the continuous noise space to refine generation quality or increase diversity. For example, Ma et al. (2025) frame inference-time scaling as a search problem over the diffusion noise space, introducing path-based search strategies that explore alternative generation trajectories and leverage a verifier to improve alignment with the target objective.

While these methods focus on continuous noise, autoregressive (AR) models enable a different form of search by directly operating over discrete token sequences. This approach is naturally compatible with AR modeling and large language model techniques, offering advantages such as finer control, easier integration with multimodal tasks, and direct token-level reasoning. Prior works like TTS-VAR (Chen et al., 2025b) have explored search within specific AR architectures. Here, we study token-level search more broadly, highlighting its potential as a general and flexible framework for improving image generation, parallel to noise-based search in diffusion models.

# B DETAILED DISCUSSION ON SEARCH ALGORITHMS & VERIFIERS

In the following we provide a more detailed breakdown of the different search algorithms we study in the experiments, as well as a discussion on the various choices of verifiers we optimize.

## B.1 SEARCH ALGORITHMS

**Best-of-$N$ sampling.** This approach generates $N$ independent samples from the AR model and then selects the one with the highest score under a verifier. It is simple and easy to parallelize, but does not make use of any potential structure in the search process, so each sample is independently generated. The alignment to the objective improves monotonically with larger $N$, but at the cost of linear growth in computation.

**Beam search.** Beam search maintains $k$ partial sequences (the "beam") at each decoding step. For each beam item, it expands using the model's top-$M$ next-token predictions, producing up to $kM$ candidates. These candidates are then scored by detokenizing the partial sequences and evaluating them with a verifier, and the best $k$ are kept for the next step. In this way, beam search explores several promising continuations rather than committing to a single greedy path. It is more structured than best-of-$N$, allowing for early guidance of the generation process. However, in the case of 2D grid tokens, early token sequences often contain little information about the final image, so early intervention may actually harm the results.

**Lookahead search.** Lookahead search goes a step further than beam search: when deciding which token to select, it not only considers the score of the current partial sequence but also performs rollouts to more complete token sequences, which can facilitate verification. For example, at the 5th token, beam search evaluates only the sequence from tokens 1–5. In contrast, lookahead search uses the AR model to generate subsequent tokens (e.g., up to 32), then detokenizes the sequence and evaluates it with a verifier. This lookahead process provides more reliable scores for each state. For 2D tokens, it is especially useful because it allows verifiers to score more complete pictures, since partial token sequences either need to be fully completed or padded to be decoded. Because of that, lookahead serach can be far more computationally expensive per search step than other search methods.

**Practical Consideration** Based on the observation that image tokens are usually dense, practically, we do not need to search at every token step but only at selected points in the sequence. We observe that generally, using more search steps leads to better performance, but this comes at the cost of increased computation time. As shown in Appendix Table 4, searching over only half of the tokens already achieves a performance close to the full search.

## B.2 VERIFIERS

**Image–text alignment.** These verifiers measure how well a generated image matches the input text, compensating for the weak control often observed in text conditioning. Several types of models can serve this role: (1) *CLIP model*, trained on large-scale image–text pairs with a contrastive loss, which directly measures semantic similarity Radford et al. (2021). (2) *Human-preference models*, such as ImageReward (Xu et al., 2023), CycleReward (Bahng et al., 2025), PickScore (Kirstain et al., 2023), and HPSv2 (Wu et al., 2023), which are trained on preference data and better capture alignment with human intent. (3) *Detection and segmentation models* use detectors (e.g., Grounded-SAM (Ren et al., 2024)) to localize objects, then apply rule-based checks to verify object presence, position, color, and count.

**Image–image alignment.** These verifiers (e.g. DreamSim (Fu et al., 2023)) measure the similarity between two images, enabling guidance not only from text, but also from reference images.

**Image quality.** These verifiers (e.g. aesthetic Score (Schuhmann et al., 2022)) evaluate a generated image independently of text, focusing on fidelity, aesthetics, or safety. They are typically fine-tuned on datasets annotated with quality labels by humans or automated metrics to predict quality scores.

**Ensemble verifiers.** We note that these verifiers are not disjoint: for example, human-preference models capture both alignment and quality. One can also combine text–image alignment with image–image alignment for multimodal control. Even within a single group, different verifiers may complement each other: object detection and semantic segmentation with rule-based scoring are more effective for verifying positions, while CLIP-like models perform better at capturing color or style. In practice, ensembling multiple verifiers often yields better results. Since different verifiers operate on different scales, we adopt a rank-based ensemble approach. Specifically, each sample is ranked according to each verifier, after which we compute the average rank across verifiers and select samples based on these aggregated ranks.

**Distillation into token-based verifiers.** All of the above verifiers operate on images, but the detokenization process from discrete tokens to images can sometimes be time-consuming. For example, FlexTok uses a flow-based detokenizer, requiring multiple denoising steps for every image. To address this, we also consider the possibility of distilling image-based verifiers to take tokens as input instead, and directly output a score. In our experiments, we test this idea with the CLIPScore verifier, see Sec. E.2 for details. This approach can also be viewed as a process reward model, which evaluates partial token sequences and predicts their final score.

## C    EXPERIMENT SETTINGS

### C.1    PRETRAINED MODELS

For FlexTok, we use the largest model `d18-28` with a pretrained AR model, and we use the same size for the 2D grid tokenizer baseline. For Janus, we use the 1B model, and for Janus-Pro, we use the 7B model. For Infinity, we use the 2B model. We keep the sampling hyperparameters at their default values and do not modify them in our experiments.

### C.2    EVALUATION SETTINGS

For the GenEval and DreamBench++ benchmarks, we follow their official evaluation code. In GenEval, we generate five images and report the score with an average of five. For experiments on COCO and DrawBench, we use verifier scores as evaluation metrics. For computation cost, we report the Number of Function Evaluations (NFE), where one generation counts as one evaluation, and the verification counts as one. For the COCO dataset, we use a subset of 300 images from Karpathy's partition.

### C.3    SEARCH SETTINGS

**Best-of-N search:** During search, we generate $N$ images by randomly sampling from the model. The resulting images are evaluated by a verifier, and the highest-scoring image is selected. In our experiments, we use $N = 50$ as the default, and vary it from 1 to 500 to study scaling behavior. The NFE for best-of-$N$ search is calculated as $N \times T$, where $T$ is the token length.

**Beam search.** Beam search conducts search during token generation. At each step, instead of sampling a single next token, we generate $C$ candidates for each of the previous beams (or tokens). A verifier assigns scores to these candidates, and we retain the best $B$ tokens, which form the beam. The process is repeated for subsequent steps. Thus, beam search has three tunable parameters: beam width $B$, candidate number $C$, and the number of tokens to search $S$. Unless otherwise specified, we use $B = 5$ and $C = 10$. For benchmark comparisons, we run a full search on all tokens; for ablation studies on different verifiers, we use a lighter setting where only a subsample of tokens is searched. Specifically, 1,2,4,8,16,32,64,128,256 in FlexTok. The NFE is calculated as $B \times T + B \times C \times S$, where $T$ is the total sequence length.

For FlexTok and Infinity, beam search results are directly detokenized. For 2D grid tokenizers such as Janus and Janus-Pro, we pad the remaining tokens with zeros before detokenization.

**Lookahead search.** Lookahead search extends beam search by evaluating several tokens ahead. We use $B = 5$ beams and $C = 10$ candidates by default. In addition to the beam width, candidate number, and search depth, lookahead introduces an extra parameter $L$, the lookahead length. For 2D grid tokenizers, we roll out until the end of the sequence. For Infinity, we find that looking ahead $L = 5$ steps is sufficient to reveal clear structure, so we use this as the default. An ablation of the lookahead parameter is reported in Table 6 and Table 6. The NFE is given by $B \times T + B \times C \times S \times (1 + L)$, where $L$ is capped by the sequence length.

## C.4 VERIFIER SETTINGS

**Implementation details for Grounded-SAM** We use grounded-SAM (Ren et al., 2024) as a spatial rule-based verifier, which combines Grounding DINO and SAM, to detect the prompt-specified objects and obtain segmentation masks. From these masks, we (i) assess object existence and counts; (ii) crop object regions for CLIP-based color classification, as in GenEval; and (iii) compute a continuous relative-position score with PoS (Rezaei et al., 2025). The final score lies in $[0, 1]$, aggregating criterion satisfaction rather than using a binary pass/fail.

**Other verifiers** The CLIPScore we used is defined as CLIPScore $= 2.5 \times \max(\text{cos\_sim}, 0)$ as introduced in (Hessel et al., 2021). For CycleReward, we adopt the Combo variant. For the other verifiers, we use HPS v2, PickScore v1, and ImageReward v1. To ensemble verifiers, following Ma et al. (2025), we rank images with each verifier, sum the ranks per image, and select the image with the best rank sum.

## D SEARCH HYPERPARAMETERS

**FlexTok + beam search** We study the three main hyperparameters in beam search, beam width, search token number, and candidate number. We conduct experiments on a COCO subset and use CLIP as the verifier. The results are shown in Table 3. For beam width experiments, we use 9 token number and 10 candidates with different beams. For the search token number, we keep the beam width to 5 and scale the selection number till selecting in each step. Then we use the full token number case and explore candidate number from 10, 20, 40. It further improves the CLIPScore. Notably, for both aesthetic and imagereward, their perofmrnace do not necessarily increase, which could indicate verifier hacking exists.

Table 3: Results of experiments with varying beam widths for conducting beam search on FlexTok. We use 10 candidates and search for 9 steps. Results shown on a 300-image COCO subset.

| Beam width ($B$) | CLIPScore | Aesthetic | ImageReward |
|---|---|---|---|
| 2 | 86.44 | 4.57 | 0.33 |
| 5 | 90.04 | 4.49 | 0.33 |
| 10 | 92.74 | 4.44 | 0.41 |
| 15 | 93.50 | 4.40 | 0.58 |
| 20 | 94.63 | 4.39 | 0.60 |
| 25 | 101.70 | 4.39 | 0.61 |

**FlexTok + lookahead search** We further study the effect of the lookahead number in Table 6. Using the default setting of beam width 5, 9 search steps, and 10 candidates, we increase the lookahead from 2 to 32 tokens. On FlexTok, we find that lookahead leads to only marginal improvements. This suggests that even without lookahead, detokenization from FlexTok's partial tokens already provides sufficient information about the token sequence.

Table 4: Results of experiments with varying token numbers for conducting beam search on Flex-Tok. We use a 5-beam width and a 10-candidate step. *For token number 9, we use tokens 1,2,4,8,16,32,64,128,256. For others, we uniformly sample from 256 tokens.

| Search token number ($S$) | CLIPScore | Aesthetic | ImageReward |
|---|---|---|---|
| 2 | 81.85 | 4.51 | 0.35 |
| 4 | 85.93 | 4.50 | 0.46 |
| 8 | 87.68 | 4.52 | 0.49 |
| 9* | 90.04 | 4.49 | 0.33 |
| 16 | 90.08 | 4.52 | 0.55 |
| 32 | 93.45 | 4.52 | 0.56 |
| 64 | 98.33 | 4.49 | 0.60 |
| 128 | 104.15 | 4.45 | 0.58 |
| 256 | 111.48 | 4.41 | 0.43 |

Table 5: Results of experiments with different candidate numbers for conducting beam search on FlexTok. We use a 5-beam width and a 256 search token number.

| Candidate number ($C$) | CLIPScore | Aesthetic | ImageReward |
|---|---|---|---|
| 10 | 111.48 | 4.41 | 0.43 |
| 20 | 114.14 | 4.41 | 0.41 |
| 40 | 116.75 | 4.35 | 0.31 |

Table 6: Results of experiments with different lookahead values for conducting lookahead search on FlexTok. We use a 5-beam width, 9-search token, 10-candidate step. Results shown on a 300-image COCO subset.

| Lookahead number ($L$) | CLIPScore | Aesthetic | ImageReward |
|---|---|---|---|
| 2 | 82.68 | 4.76 | 1.03 |
| 8 | 82.30 | 4.75 | 1.08 |
| 32 | 83.18 | 4.76 | 1.14 |

**Janus + beam search and lookahead search**  In a setting similar to FlexTok, we study the effects of varying beam widths and search token numbers in Janus. As shown in Table 7, increasing either parameter improves performance. Table 8 further shows that increasing the lookahead number leads to more efficient gains. We set candidate number to 10 for all these experiments.

Table 7: Results of varying beam width ($B$) with $S = 9$ (left) and varying search token number ($S$) with $B = 5$ (right) in Janus + beam search.

| Beam width ($B$, $S = 9$) | | | | Search token number ($S$, $B = 5$) | | | |
|---|---|---|---|---|---|---|---|
| $B$ | CLIPScore | Aesthetic | ImageReward | $S$ | CLIPScore | Aesthetic | ImageReward |
| 2 | 85.37 | 4.83 | 0.21 | 4 | 87.49 | 4.80 | 0.24 |
| 5 | 87.59 | 4.80 | 0.15 | 9 | 87.59 | 4.80 | 0.15 |
| 10 | 89.02 | 4.79 | 0.27 | 73 | 88.19 | 4.81 | 0.17 |
| | | | | 576 | 92.18 | 4.61 | -0.19 |

# E  ADDITIONAL VERIFIER RESULTS

## E.1  LIKELIHOOD VERIFIER RESULTS

Table 8: Results on different lookahead numbers in Janus + lookahead search. (beam width = 5, search token number = 9)

| Lookahead | CLIPScore | Aesthetic | ImageReward |
|---|---|---|---|
| 8 | 88.21 | 4.81 | 0.29 |
| 64 | 90.58 | 4.83 | 0.33 |
| 128 | 91.49 | 4.83 | 0.26 |
| All | 92.27 | 4.84 | 0.38 |

We can use the likelihood of a token sequence, $p(x)$, as a verifier. Unlike all other verifiers, the likelihood verifier enjoys no verification overhead, since the likelihood is already computed during sampling. It is also intrinsic to the AR model. For the results in Table 9, we do search on FlexTok with the standard beam search setting. Search on the likelihood verifier provides a slight boost to most metrics, though less than search with a separate verifier such as CLIPScore or ImageReward.

Table 9: Results of the likelihood verifier. We report other verifier scores and the overall GenEval performance.

| Method | CLIP | ImageReward | Aesthetic | CycleReward | HPSv2 | Spatial | PickScore | GenEval(Overall) |
|---|---|---|---|---|---|---|---|---|
| FlexTok | 31.98 | 0.543 | 4.617 | 0.636 | 0.249 | 0.660 | 0.196 | 57 |
| FlexTok + SoT, CLIP verifier | 35.66 | 0.8171 | 4.6769 | 0.9486 | 0.2551 | 0.6918 | 0.249 | 63 |
| FlexTok + SoT, ImageReward verifier | 32.72 | 1.2944 | 4.7776 | 1.0978 | 0.2640 | 0.7048 | 0.277 | 67 |
| FlexTok + SoT, Likelihood verifier | 32.34 | 0.647 | 4.749 | 0.796 | 0.258 | 0.703 | 0.277 | 60 |

### E.2 TOKEN-BASED VERIFIER

When performing search over tokenizers that use diffusion detokenizers, such as FlexTok, the detokenization is easily the most expensive part of SoT. Diffusion is far more expensive than either the AR token generation or the verifier itself. Inspired by Becker et al. (2025), we train a tiny verifier model that scores tokens directly on CLIPScore. While the token-based verifier is not completely aligned with the original model, using it as the verifier in SoT still produces sizeable gains on the original CLIPScore at a significantly reduced search cost. We get around a significant portion of the benefit of full search, with the search time itself reduced by a factor of around 200.

Table 10: Search with token-based verifier.

| Verifier | CLIPScore | Search Overhead (sec.) |
|---|---|---|
| None | 78.5 | 0.0 |
| CLIP | 95.5 | 571 |
| OpenCLIP | 85.5 | 571 |
| Token-based verifier | 85.6 | 2.6 |

### E.3 COCO VERIFIER ABLATIONS

We show additional COCO verifier ablation results in Figure 9.

## F ADDITIONAL COMPARISONS

### F.1 SEARCH OVER TOKENS VS. SEARCH OVER DIFFUSION NOISE

A primary existing visual search method is searching over initial noises in a diffusion model, as opposed to searching over tokens. We provide comparison between SoT and a representative method, Ma et al. (2025). In Table 11 and Figure 10, we provide a comparison of scaling behavior with

Figure 9: COCO Verifier Ablations. Each row reports search using one verifier. All methods use the same beam search algorithm on FlexTok. The best score in each column is highlighted in bold. The superscript in each cell represents the rank within that column's metric, and the last column reports the average of these column-wise ranks, providing an overall rank for each verifier.

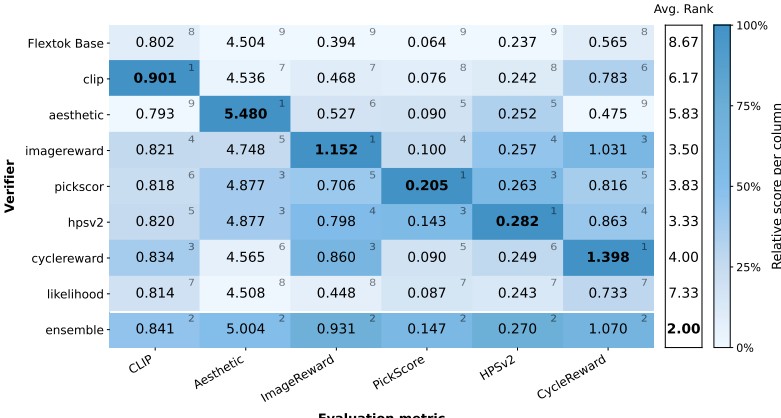

Figure 10: CLIPScore on best-of-N Flux.1-dev, a diffusion model, vs. FlexTok + `SoT`, which is autoregressive.

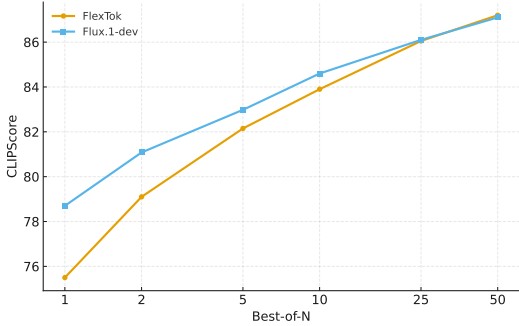

best-of-N sampling between Flux.1-dev, the diffusion model used in Ma et al. (2025) and FlexTok. Both types of search consistently improve model performance and get better with increased scale.

Table 11: Search over tokens vs. search over diffusion noise. Performance is evaluated on Draw-Bench per Ma et al. (2025).

| Method | Search on CLIP | Search on ImageReward |
|---|---|---|
| Flux | 79.3 | 0.99 |
| Flux, best-of-5 | 83.1 | 1.34 |
| Flux, best-of-50 | 87.0 | 1.54 |
| FlexTok | 76.4 | 0.10 |
| FlexTok, best-of-5 | 82.2 | 0.77 |
| FlexTok, best-of-50 | 87.2 | 1.16 |

# G ADDITIONAL VISUALIZATION RESULTS

## G.1 DPG BENCHMARK

We show generation results on the DPG benchmark comparing vanilla Janus-Pro with our method in Figure 11.

## G.2 DreamBench++

We show additional generation results on the DreamBench++ in Figure 12.

## G.3 Visual comparison for different verifiers

We show the visual difference of search using different verifiers in Figure 13.

# H Additional AR prior results

We examine the role of the prior model during search. The main utility of the AR prior is to propose high likelihood token candidates, which drastically reduces the search space that needs to be verified. That said, the coarse-to-fine structure of certain tokenizers like FlexTok makes them amenable to explore the question: Is it possible to generate images without any AR prior, and purely through search? To do so, we perform beam search with a uniform prior, which assigns the same probability to each candidate token. In addition, we evaluate an unconditional prior model, where we use the FlexTok AR model without text conditioning. For both the uniform prior and unconditional search experiments, we uniformly subsample 1% of the total tokens (640 tokens). Compared to our default setting, which verifies around 10 candidate tokens from the AR prior, this approach is much more costly at each search step, so we only search for a 32-token sequence rather than a full 256-token sequence, and also compare with conditional priors and no search AR with the same 32-token sequence length.

To measure the generation quantitatively, we take a subset of GenEval containing 180 prompts measuring single-object and two-object generation. Using pure search, we achieve about 79% accuracy on single-object generation and 32% on two-object generation, which is reasonable. Unconditional AR prior further improves performance, while conditional AR achieves the highest accuracy. We show visualizations of comparing the uniform and unconditional AR priors with a standard text-conditional AR prior in Fig. 14. We find that search without an AR prior is indeed possible and yields reasonable results (Fig. 14, top row). Both unconditional and conditional priors, however, effectively narrow the search space towards more likely next tokens, enabling more efficient exploration.

Table 12: **Quantitative comparison of three different priors for search.** We compare the performance of a uniform prior, an unconditional AR prior, and a conditional AR prior using beam search on the GenEval subset with FlexTok. Generation through search alone is possible; however, stronger priors lead to better performance. Nonetheless, search-only generation still lags behind AR-only generation, underscoring the importance of the prior model. Results in Acc. (%).

| Prior | Search | Single Object | Two Object |
|---|---|---|---|
| Uniform Prior | yes | 79 | 32 |
| Unconditional AR | yes | 85 | 33 |
| Conditional AR | yes | **100** | **81** |
| Conditional AR | no | 97 | 48 |

# I LLM Usage

We used ChatGPT in the following ways: (1) To improve the writing: editing grammar, style, and clarity. (2) To help explain ideas more clearly (rewriting sentences, smoothing language). (3) To assist with literature retrieval: suggesting related works and finding papers during background and related-work writing. (4) To generate demonstration images in Fig 1 (a) and to generate the code for plotting the Flux vs. FlexTok comparison in Fig. 10. All outputs from the LLM were checked, verified, and revised by the authors. The authors remain fully responsible for the final content.

1188
1189
1190
1191
1192
1193
1194
1195
1196
1197
1198
1199
1200
1201
1202
1203
1204
1205
1206
1207
1208
1209
1210
1211
1212
1213
1214
1215
1216
1217
1218
1219
1220
1221
1222
1223
1224
1225
1226
1227
1228
1229
1230
1231
1232
1233
1234
1235
1236
1237
1238
1239
1240
1241

Generated by Janus-Pro          Generated by Janus-Pro + SoT (Ours)

A contented sloth, with a wide grin on its face, is decked out in an eclectic ensemble featuring a sleek black leather jacket and a brown cowboy hat atop its head. It's also sporting a traditional tartan kilt paired with a smart red bowtie around its neck. **In one claw,** the sloth firmly grips a wooden quarterstaff, **while the other supports a large, thick book** with a leather-bound cover.

A festive array of red and yellow balloons tied with curling ribbons, gently bobbing from the breeze of a spinning ceiling fan. **The fan has wooden blades and a brass finish,** which contrasts with the bright colors of the balloons. The balloons are clustered in a joyful bunch, casting soft shadows on the ceiling above.

A steaming bowl of Pho, with a rich, clear broth and a generous topping of fresh bean sprouts. The bowl sits on a dark wooden table, accompanied by **a side plate of lime wedges and basil leaves**. The noodles are submerged beneath the broth, and thin slices of beef float on the surface, partially obscured by the green sprouts.

Two ceramic cups filled with steaming coffee are placed on a wooden table with a natural grain finish. **The cup on the left showcases intricate latte art spelling out the word "LOVE" with a heart-shaped design**, while the cup on the right has the word "PEACE" beautifully crafted atop its frothy surface. Both cups have a glossy finish, and the warm lighting accentuates the creamy texture of the latte art.

An intricate Chinese ink and wash painting that depicts a majestic tiger, its fur rendered in delicate brush strokes, wearing **a traditional train conductor's hat atop its head**. The tiger's piercing eyes gaze forward as it firmly grasps a skateboard, which features a prominent yin-yang symbol in its design, symbolizing balance. The background of the painting is a subtle wash of grays, suggesting a misty and timeless landscape.

Figure 11: **Example prompts from the DPG benchmark (Hu et al., 2024) and the corresponding generated images from Janus-Pro (Chen et al., 2025a) and ours (enhanced by search).** Here, we use best-of-N (N=50) sampling with the imagereward verifier.

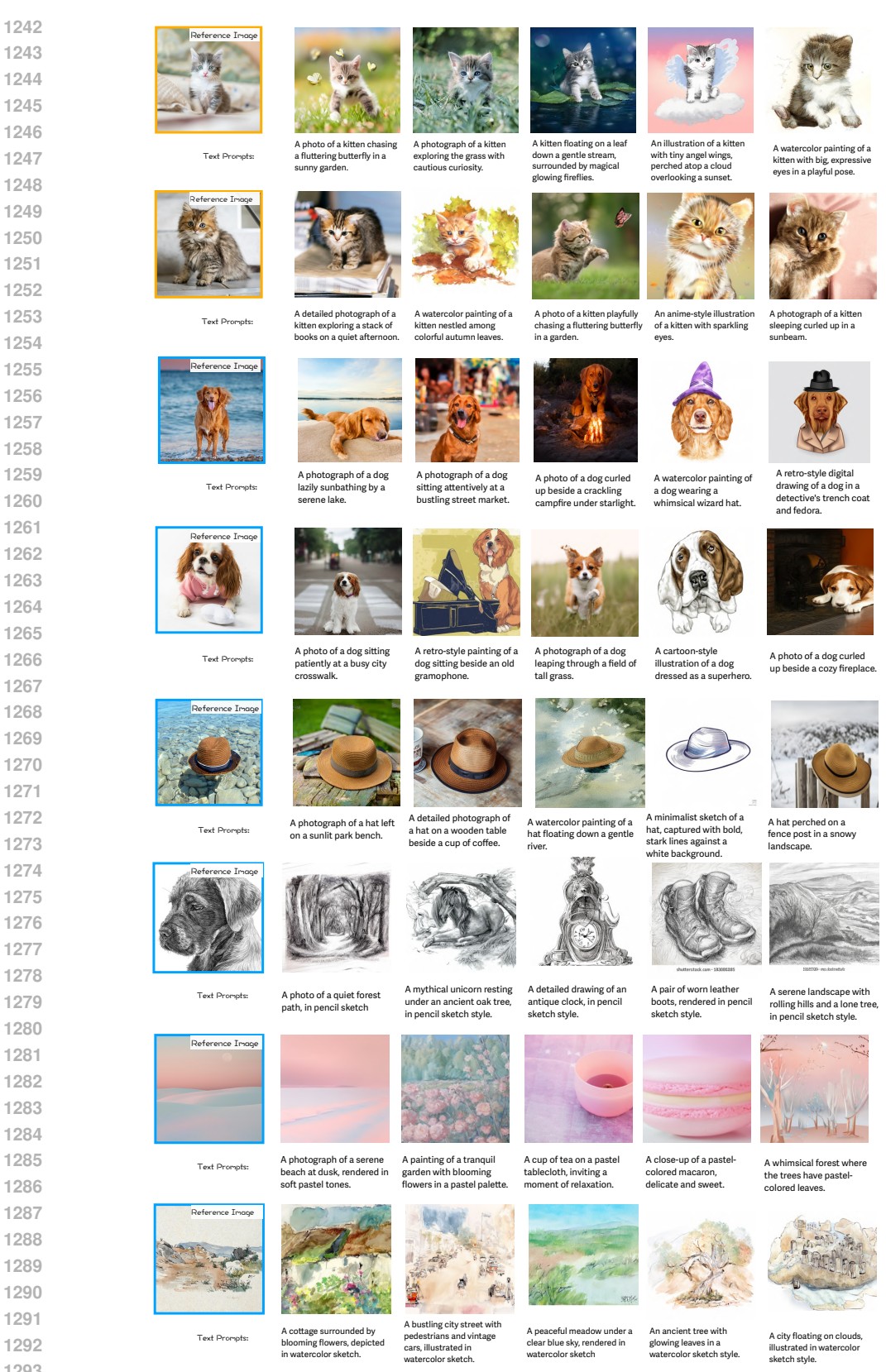

Figure 12: **Example reference images and prompts from DreamBench++ (Peng et al., 2024)**: Images are generated by FlexTok (Bachmann et al., 2025) with Beam search, using DreamSim (Fu et al., 2023) as verifier.

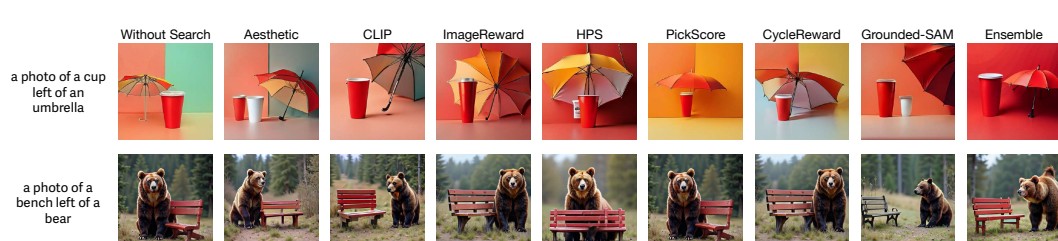

Figure 13: **Search with different verifiers**: We show the visualization comparison of the best-of-50 search results on Janus-Pro guided by different verifiers on the GenEval benchmark.

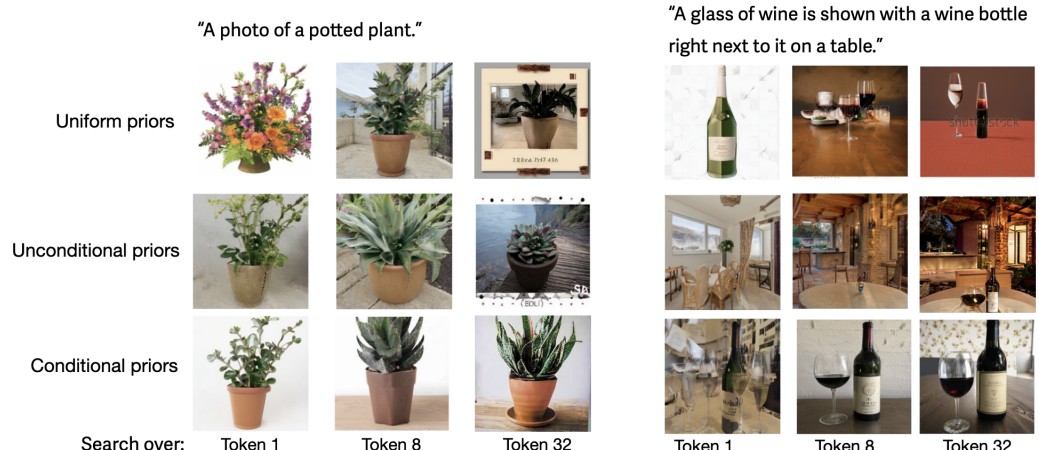

Figure 14: **Visual comparison of three different priors for search.** We evaluate: (1) *a uniform prior* that samples randomly from the entire token space, (2) *an unconditional prior* that uses an unconditional AR model to guide generation, and (3) *a conditional prior* that uses a text-conditional AR model to predict the probability of the next token. We find that while generation by pure search is possible, a well-trained AR model greatly reduces the search space and makes generation more efficient and more robust to verifier hacking.

