# OpenReview forum: "Generation by Search: Scaling Test-Time Compute for Autoregressive Image Generation"
_ICLR.cc/2026/Conference — ICLR 2026 Conference Withdrawn Submission_

### Official Review · Reviewer_s9CM · 2025-10-21

**Soundness:** 3
**Presentation:** 2
**Contribution:** 1
**Rating:** 4
**Confidence:** 5

**Summary:**

The paper proposes a test-time search framework for autoregressive image generation that treats generation as navigating a token-level search tree guided by verifier objectives. It systematically studies four axes, including token structure, search algorithm, verifier choice, and the AR prior. Across multiple AR models and benchmarks, SoT consistently improves alignment and controllability, exhibits favorable test-time scaling, enables zero-shot multimodal control via image verifiers, and shows that ordered tokenizations can even work with weak or no AR priors. The paper also analyzes verifier hacking risks and reports that ensembles and human-preference verifiers are more robust, providing practical guidance for deploying test-time search in AR image generation.

**Strengths:**

* The paper reframes autoregressive image generation as a search problem, unifying it with LLM-style test-time search.
* It conducts a systematic study of test-time scaling with thorough experiments across verifiers, AR priors, and search algorithms.
* The paper is overall simple, clear, and easy to follow.

**Weaknesses:**

The core idea—applying test-time scaling to image generation—is straightforward and largely mirrors TTS in LLMs (e.g., beam search, best-of-N), with limited methodological novelty. The paper also omits key prior work in visual generation: for example, [1] as an early AR image-generation TTS method, and [2,3] as diffusion-based TTS approaches. Compared to these works, this paper contributes neither a new algorithmic technique nor fresh insights. For instance, while the authors discuss verifier hacking in TTS, they do not offer concrete mitigation strategies.

The experiments lack fair wall-clock comparisons. Reporting only NFE/FLOPs obscures practical deployment costs; comprehensive latency/throughput measurements (including detokenization overhead and verifier run-time) are needed for an apples-to-apples assessment.


[1] Guo, Ziyu, et al. "Can We Generate Images with CoT? Let's Verify and Reinforce Image Generation Step by Step." arXiv preprint arXiv:2501.13926 (2025).

[2] Zhuo, Le, et al. "From reflection to perfection: Scaling inference-time optimization for text-to-image diffusion models via reflection tuning." arXiv preprint arXiv:2504.16080 (2025).

[3] Li, Shufan, et al. "Reflect-DiT: Inference-Time Scaling for Text-to-Image Diffusion Transformers via In-Context Reflection." arXiv preprint arXiv:2503.12271 (2025).

**Questions:**

What exactly is the rollout strategy used in Figure 2? For example, in Janus, when only a small number of tokens have been generated early on, what does the detokenized image look like? Can the verifier reliably evaluate such partially decoded images?

---

### Official Review · Reviewer_NgrF · 2025-11-01

**Soundness:** 2
**Presentation:** 3
**Contribution:** 2
**Rating:** 2
**Confidence:** 4

**Summary:**

This paper implements the test-time scaling algorithm for autoregressive (AR) image generation and adopts a tree-based search strategy that better aligns with the characteristics of AR models. The approach leads to a measurable performance improvement, supported by extensive experimental validation.

**Strengths:**

The validation experiments are extensive and convincing, complemented by clear and informative figures and tables that make the results easy to interpret.

**Weaknesses:**

1. The paper lacks clear novelty. It explores a well-known training-free approach for autoregressive (AR) image generation, but the proposed method is highly similar to existing works and does not offer distinctive innovations.
2. As shown in Figure 2, decoding begins before the full image prediction is completed. This raises a concern: could such partial decoding introduce errors or instability in the verifier’s evaluation?
3.Section 3.2 claims that the method follows a “coarse-to-fine” generation process consistent with human visual perception. However, true coarse-to-fine generation should occur at a global image level (as in diffusion models). Since this work only decodes partial tokens, it remains unclear how the claimed coarse-to-fine property is achieved.
4.It is uncertain whether TTS provides consistent benefits throughout the entire generation process. For example, when predicting the second token, the model only conditions on the first token’s TTS result—does this provide meaningful guidance? Has the paper explored the possibility of selectively applying TTS to certain stages or tokens?

**Questions:**

Please provide responses to the issues raised in my Weaknesses section.

---

### Official Review · Reviewer_mjcm · 2025-11-01

**Soundness:** 3
**Presentation:** 3
**Contribution:** 2
**Rating:** 4
**Confidence:** 4

**Summary:**

This paper proposes a test-time scaling framework for autoregressive (AR) image generation, formulating the generation process as a token-level search guided by verifiers. The method, termed Search over Tokens (SoT), explores multiple search algorithms (e.g., best-of-N, beam, lookahead) and verifier types to optimize image-text alignment and visual quality without retraining. Extensive experiments across benchmarks (GenEval, COCO, DreamBench++) demonstrate consistent improvements in alignment and controllability across various AR architectures. The study systematically analyzes four design axes—token structure, search algorithm, verifier, and AR prior—establishing test-time search as a scalable and general paradigm for enhancing AR image generation.

**Strengths:**

1. The proposed approach is conceptually clear and easy to follow, making it accessible to readers familiar with autoregressive image generation.
2. The experimental design is well thought out, with a logical structure that effectively supports the paper’s main claims.

**Weaknesses:**

1. The contributions are limited. Although the introduction lists three claimed contributions, in practice only the first one—the design of a test-time scaling (TTS) framework—constitutes a substantive contribution. The second and third points (related to model choice, verifier design, search algorithm, and token structure) are more like implementation necessities rather than conceptual advances.
2. In the introduction, when motivating the use of TTS, the authors only reference applications of TTS in large language models (LLMs). However, the idea of TTS was originally introduced and popularized in diffusion-based image generation methods, which should have been properly acknowledged.
3. Diffusion-based TTS algorithms typically evaluate the entire image, whereas the proposed approach evaluates partially decoded images based on subsets of tokens. It seems that the former formulation might be more principled and semantically meaningful.
4. The paper provides very limited discussion of the verifier component, which should be a central aspect of the design. It remains unclear how the verifier effectively assesses the impact of each partially decoded token on the global image quality, and how this aligns with the TTS objective.

**Questions:**

Please refer to the "Weakness".

---

### Official Review · Reviewer_Uvri · 2025-11-07

**Soundness:** 2
**Presentation:** 2
**Contribution:** 2
**Rating:** 4
**Confidence:** 2

**Summary:**

This paper reformulates autoregressive image generation as a token-level search problem and introduces the Search over Tokens (SoT) framework. SoT treats the pretrained AR model as a probabilistic prior and searches for token sequences that maximize verifier-based rewards such as CLIPScore and ImageReward. The framework systematically explores the design space across four axes: token structure, search algorithm, verifier type, and prior usage.

**Strengths:**

- The paper provides a unified framework for test-time search in autoregressive image generation, connecting recent progress in inference-time scaling with structured token-level optimization.

- It systematically evaluates a broad design space across tokenization, search strategy, and verifier composition, revealing empirical trends that can guide practical deployment.

- The results show clear and reproducible scaling behavior under increased compute budgets, reinforcing the value of search-based generation as a general paradigm for AR models.

**Weaknesses:**

- The contribution is largely empirical and heuristic, focusing on organizing and benchmarking existing components such as search algorithms, verifiers, and AR priors rather than introducing new theoretical insights or algorithmic principles.

- The framework’s performance depends heavily on specific verifiers such as CLIP and ImageReward, but the paper offers limited analysis of verifier bias, reward overfitting, or robustness to unseen evaluation metrics.

- Comparisons to other recent search-based or test-time scaling approaches including speculative decoding and inference-time optimization in diffusion or LLMs remain superficial, leaving unclear whether SoT offers fundamentally new capabilities beyond unification and benchmarking.

**Questions:**

- How robust are SoT’s improvements under alternative verifiers or human evaluation when the optimization target differs from CLIP or ImageReward?

- Can the authors clarify how SoT conceptually differs from other inference-time scaling or search-based generation frameworks, and whether its findings generalize beyond benchmark-specific settings?

- Have the authors analyzed whether increasing search depth or relaxing the AR prior leads to diminishing returns or mode collapse, similar to behaviors observed in recent reward-based inference studies?

---

### Note · Authors · 2025-11-14

I have read and agree with the venue's withdrawal policy on behalf of myself and my co-authors.